# FEW-SHOT PARAPHRASE GENERATION WITH LLMS: AN EMPIRICAL STUDY OF MODELS AND HYPERPARAMETERS

## ABSTRACT

The rapid progress of Large Language Models (LLMs) has made them widely used for data augmentation tasks, notably through paraphrase generation. For that task, paraphrases are expected to preserve the original meaning while exhibiting lexical diversity.

In this work, we conduct an empirical study of various off-the-shelf LLMs for paraphrase generation. We examine different prompting and decoding strategies, and compare systems with respect to their ability to follow predefined templates, retain semantic fidelity, and produce lexical variation.

Our results show that LLMs are generally effective at generating paraphrases. However, guiding the generation process by providing initial tokens significantly improves adherence to required patterns. Under this condition, repetition penalties in decoding can further enhance output diversity. Interestingly, we also find that few-shot prompting may reduce lexical diversity.

## 1 INTRODUCTION

Paraphrase generation is an important task in natural language processing. The ability to produce sentences that are lexically different from the original while preserving their meaning enables effective data augmentation, which has been widely studied and applied (Kumar et al., 2019; Gao et al., 2020; Dai et al., 2023). This has improved the performance of several downstream applications, such as question Dong et al. (2017), machine translation Mehdizadeh Seraj et al. (2015); Yang et al. (2019), text simplification Maddela et al. (2021), and text revision Raheja et al. (2024).

In this work, we focus on the ability of language models to generate paraphrases that are lexically distant from the source sentences. Recent studies have explored the paraphrasing capabilities of large language models (LLMs), analyzing their adherence to instructions and providing qualitative evaluations on models such as GPT-4 (Chataigner et al., 2025; Meier et al., 2025; Vahtola et al., 2025). However, to our knowledge, no prior work has specifically investigated paraphrasing for the purpose of maximizing lexical diversity by taking advantage decoding penalties in a few-shot setting without sampling.

To measure lexical differences at the sentence level, we use two metrics: the Levenshtein distance (edit distance), which counts the minimum number of deletions, insertions and replacements of characters, required to transform one sentence into another, and the Jaccard distance between word stems of candidate and source sentences.

We evaluate both medium sized instructed models and smaller models fine-tuned on paraphrase corpora. Our experiments cover 4 datasets for a total of 1,456 original sentences, chosen because they were recently published and are therefore unlikely to have been included in the training data of the evaluated models. For fairness, all models were tested with the same prompts and decoding parameters. However, since small models are not instructed, prompts were not provided to them.

Our main findings can be summarized as follows: (1) Providing initial tokens to guide generation substantially improves adherence output scheme. (2) Applying repetition penalties during decoding consistently enhances lexical diversity. (3) Few-shot prompting, contrary to expectations, often reduces diversity instead of improving it.

The remainder of this paper is organized as follows: Section 2 introduces the models, prompting strategies, and decoding parameters considered. Section 3 describes our experimental protocol. Section 4 presents and analyzes the results, and Section 5 concludes with key findings and future research directions.

## 2 PARAPHRASE GENERATION USING LLM

Recent advances in large language models (LLMs) have made them particularly effective for paraphrase generation. Instructed LLMs can generate paraphrases in zero-shot or few-shot setups and can be guided to satisfy specific stylistic or structural constraints. In this work, we take an empirical approach to understand how paraphrase generation is affected by the choice of model, the prompting strategy used to present the task, and the decoding parameters that control token selection. The following subsections detail these aspects.

### 2.1 MODELS CONSIDERED

We evaluate a range of off-the-shelf models for paraphrase generation, grouped into two categories: Non-instruct models and Instruct models.

**Non-instruct models:** These models, or fine-tuned variants, are explicitly trained for paraphrasing using aligned examples. They are lightweight (0.2-0.4B parameters) and rely on direct fine-tuning rather than large-scale instruction alignment. Such models generate a paraphrase from a source sentence without requiring an explicit instruction. However, this lack of instruction handling makes it difficult to impose constraints during generation. They may tend to reproduce patterns seen in their fine-tuning data. This group includes BART-based and T5-based systems (Raffel et al., 2023), as well as Parrot.

**Instruct models:** Instruction-tuned LLMs are trained on massive corpora with the goal of mimicking human answers and reasoning across a wide range of tasks. They exhibit strong zero- and few-shot performance and can be guided through specific prompts to generate paraphrases according to task-specific constraints. However, previous studies have shown that the prompting strategy strongly affects performance (Brown et al., 2020). A notable challenge with these models is their tendency to be verbose. They often produce long explanations (Poddar et al., 2025; Zheng et al., 2023b) or reasoning chains (Xu et al., 2025), which complicates the extraction of candidate paraphrases.

We evaluate multiple instruct model families ranging from 3–8B parameters, including Llama 2&3, Mistralv0.2&3, Qwen2.5, Mamba, and Falcon-Mamba. Almost all instruct models follow the decoder-only Transformer architecture (Vaswani et al., 2017), except Mamba and Falcon-Mamba, which use the Mamba architecture (Gu & Dao, 2024b).

Table 1 summarizes their main characteristics, including parameter count and vocabulary size. Vocabulary size is relevant since lexical diversity may partly depend on the richness of the token inventory. This diverse set allows us to compare dedicated paraphrase generation systems against general-purpose instruction-tuned LLMs, and to analyze the effect of model scale and training paradigm on paraphrase quality.

Table 1: Considered models to generate paraphrases and their main characteristics.

| Type | Name | Params. (B) | Voc. (K) | Reference |
|:---:|:---:|:---:|:---:|:---:|
| Not instruct | Bart-paraphraser | 0.22 | 50 | Lewis et al. (2019) |
| | Parrot | 0.41 | 32 | Damodaran (2021) |
| | T5-paraphraser | 0.22 | 32 | available online[1] |
| | T5-chatGPT-paraphraser | 0.22 | 32 | Vladimir Vorobev (2023) |
| Instruct | Llama2 | 6.74 | 32 | Touvron et al. (2023) |
| | Llama3 | 8.03 | 128 | Grattafiori et al. (2024) |
| | Mistralv0.2 | 7.24 | 32 | Jiang et al. (2023) |
| | Mistralv0.3 | 7.25 | 33 | Jiang et al. (2023) |
| | Qwen2.5 | 7.62 | 152 | Bai et al. (2023) |
| | Mamba | 2.77 | 50 | Gu & Dao (2024a) |
| | Falcon-mamba | 7.27 | 65 | Zuo et al. (2024) |

---

[1]https://huggingface.co/ramsrigouthamg/t5_paraphraser

Table 2: Considered prompts to generate paraphrases and their main characteristics. Prompts considered, 0.S and F.S designates respectively one-shot and four-shots examples.

| Shots | | Source | Paraphrase |
|---|---|---|---|
| | | **Standard** | |
| F.S | 0.S | The little cat refreshes himself with water every morning. | Every morning, the little cat refreshes himself by drinking water. |
| | | There is a big tree in my garden. | A great tree is planted in my garden. |
| | | Some kids are more adventurous than others. | Some children are less afraid of the unknown than others. |
| | | I love my previous car! | My old automobile is still in my heart. |
| | | **Vulgar** | |
| F.S | 0.S | I stepped on a piece of shit this morning. | I was out for a walk this morning when I trample on a poop. |
| | | Fuck both of you. | Go fuck yourselves. |
| | | Don't listen to him, he's batshit crazy. | Fucking ignore that wanker, he's fucking mental! |
| | | That fuckin' heap of junk ain't worth shit, it won't fuckin' run! | That goddamn car is utter crap, it refuses to start. |

## 2.2 PROMPTING STRATEGIES

The performance of instruction-tuned LLMs is strongly influenced by how tasks are presented, making the choice of prompting strategy crucial for paraphrase generation (White et al., 2023). We study prompting paradigms along two axes: the number and type of examples provided in the prompt, and whether the generation is guided or free-form.

**Examples in the prompt:** Prompts can be configured in zero-shot or few-shot setups. In zero-shot prompting, the model receives only a task instruction without any input–output examples. This tests the model's ability to generalize directly from pre-training and instruction-tuning. In few-shot prompting, the instruction is augmented with a small number of input–output examples (1 or 4 in our study). These examples illustrate the desired style, lexical variation, or structural patterns of the target paraphrases. Few-shot prompting can guide task-specific behavior, but may reduce diversity if the examples are repetitive or restrictive. To further study model sensitivity, we consider two types of examples: a standard set and a set containing vulgar expressions. Since many on-the-shelf models are tuned to avoid sensitive topics using RLHF (Zheng et al., 2023a), one may wonder whether adding vulgar examples to the prompt could unlock such responses or, conversely, constrain them. These examples are summarized in Table 2.

We consider two strategies for controlling the model output: (1) **Free-text generation**: the model generates a paraphrase freely in a chat-like manner ; (2) **Continue generation with initial tokens**: the model receives the beginning of the target template, guiding the format. This improves adherence to predefined patterns needed to automatically extract the generated paraphrase. Overall, we define 10 prompting templates combining the number of shots (0, 1, or 4), the continuous generation mode (C), and the example type (standard or vulgar, V). For instance, "1S.V.C." indicates one-shot prompting with a vulgar example set and continuous generation. These templates are summarized in Table 3. Models that are not instruction-tuned do not require a prompt and are treated as zero-shot (0S) only. These prompting strategies define the input context for the model and interact closely with decoding parameters, which are discussed in the next subsection.

## 2.3 DECODING PARAMETERS

Modern language models are autoregressive: given a context, they predict the next subword by assigning a probability to each token in their vocabulary. There exist several strategies for selecting the next token from this probability distribution (Shi et al., 2024). The most common are greedy decoding, sampling, and beam search.

**Greedy decoding**: selects the token with the highest probability at each step. **Sampling**: randomly selects a token according to the probability distribution, introducing stochasticity. **Beam search**: explores multiple candidate sequences in parallel, considering whether lower-probability tokens may lead to more relevant overall generations. Beam search can be combined with either greedy or sampling strategies to control

Table 3: Prompt templates used for the considered strategies.

| Prompt | Variant |
|---|---|
| (*user*): *You have to transform a sentence 1) into a paraphrase 2). The purpose of 1) to 2) transformation is to maintain the original meaning of the sentence 1) in 2). You must respect a constraint.* (*assistant*): *I've understood the instructions perfectly. My answer will follow the following format 2) " "* | |
| (*user*): *1) "{example sentence}" <CONSTRAINT> "Use words that are as different as possible from the original sentence." < / CONSTRAINT>* (*assistant*): *2) "{example paraphrase}"* | Repeated X times for X-shot prompting. Examples can be standard or vulgar. |
| (*user*): *1) "{sentence to paraphrase}"* | |
| (*assistant*): *2) "* | Only for continue generation. |

randomness during exploration. Additional mechanisms can penalize certain tokens before selection. A diversity penalty discourages candidate tokens that have already appeared in other beams. A penalty of 0.0 corresponds to no penalty. The no-repeat n-gram constraint prevents the model from repeating n-grams of a fixed size: if an n-gram already appears in the generated sequence, it cannot be selected again. A value of 0 allows all n-grams to be repeated.

In our experiments, we evaluate three decoding configurations, summarized in Table 4. To ensure reproducibility, we do not use sampling and instead rely on greedy decoding as the base strategy. To limit computational cost, we set a maximum generation length of 100 tokens; generation stops if this limit is reached. The "small penalty" and "huge penalty" configurations correspond to the settings provided by the authors of Parrot and T5-ChatGPT-paraphraser models. These were designed to encourage the generation of tokens that diverge from the source sentence, thereby avoiding simple copying in paraphrasing tasks.

Table 4: Decoding parameters used in our experiments

|  | No penalty | Small penalty | Huge penalty |
|---|---|---|---|
| **Strategy** | greedy | greedy | greedy |
| **Beam size** | 1 | 5 | 5 |
| **max new tokens** | 100 | 100 | 100 |
| **diversity penalty** | 0.0 | 2.0 | 3.0 |
| **no repeat gramm size** | 0 | 0 | 2 |

## 3 EXPERIMENTAL SETUP

The experiment aims to evaluate the ability of language models to generate paraphrases that are lexically distinct from the original sentence. We use 3 evaluation criteria, the sentence production rate, the meaning preservation and the diversity. Each instruct model paraphrases a sentence 10 times, for each prompting template, and this is done 3 times, for each decoding parameter. Each non instruct model paraphrases a sentence 3 times, for each decoding parameter. Thus, for each sentence, 222 generations are made. Each generation is first evaluated with a regular expression to ensure that it follows the output scheme describe in the prompt. If it's the case it is then automatically labeled as paraphrase or non paraphrase. First we introduce corpus of sentences that we will try to paraphrase. Then we introduce our evaluation metric in followings subsections.

### 3.1 CORPORA

As the majority of state-of-arts LLM are trained on all datasets available on internet, a contamination bias is likely to occur during an evaluation of generation capacities. To minimize this bias as much as possible, in this experiment we use small recently published datasets that are relatively unknown to the community. We used 4 datasets originally designed for paraphrasing. They consist of sentence pairs labeled by humans as paraphrases or non-paraphrases. In this experiment we consider all sentences individually as sources sentences. We cleaned these datasets by removing any duplicate sentences.

Table 5: Corpora characteristics

| Name | Size in sentences | Len in words | S-Bleu |
|------|-------------------|--------------|--------|
| HC-S | 197 | $10.26_{\pm0.64}$ | $0.27_{\pm0.04}$ |
| HC-Q | 198 | $8.23_{\pm0.39}$ | $0.22_{\pm0.04}$ |
| LLM | 777 | $22.48_{\pm0.43}$ | $0.45_{\pm0.02}$ |
| MCPG | 284 | $13.19_{\pm0.81}$ | $0.47_{\pm0.04}$ |

**HC dataset:** The HC dataset (Lemesle et al., 2025) was manually created. It contains short sentences and questions. We split this dataset into two subsets, HC-S that contain all sentences and HC-Q that contain all questions. Trying to paraphrases questions is particularly interesting in our case. Indeed it would not be surprising if instructed models forgot the task requested in the prompt to answer the question instead of paraphrasing it. Here is a typical example of sentence and question in this dataset: "*He can't take the joke.*" and "*How does it look to have a bun in the oven?*"

**LLM dataset:** The LLM corpus (Lemesle et al., 2025) has been generated automatically thanks to Mistralv0.2 and LLama2 models. Sources sentences were randomly picked from two well known paraphrases corpus, PAWS (Yuan et al., 2019) and MRPC (Dolan et al., 2005). Here is an example of sentence from this set: "*Trading volume was incredibly light at 500.22 million shares, below an already thin 611.45 million exchanged at the same point Thursday.*". It is important to note that, as source sentences of the original corpus came from well known corpora, they are likely to be present in training set of models. Nevertheless it is not the case for their generated paraphrases.

**MCPG dataset:** MCPG set (Fabre et al., 2021) has been generated by Monte-Carlo Tree Search algorithm using paraphrase tables with pivot language. Here is common example of sentences of this dataset: "*a very old and rusted train parked on the tracks.*". Part of source sentences from the original corpus came from COCO corpora (Lin et al., 2015) and are likely to be present in training set of studied models.

In the end all the data used for this experiment have been reviewed by humans, which offers partial guarantee of their syntactic quality. By looking at Table 5 you can see that the LLM sets is the most important as it contains $0.53\%$ of the data used. You can also see that sentences in this corpus are significantly longer as they contain, on average, roughly twice as many words as the others datasets. Let's take a look on the Self-Bleu Zhu et al. (2018) (S-Bleu) column which describe the mean n-gramm overlap of each sentences of the corpus in regards to it-self. We can observe that sentences in MCPG and LLM sets are similar, which is less true in the case of HC set. We hope that these corpora provide us a sufficiently wide view of different sentences types, as their sentence lengths are different and their contain sentences generated by human, language model and statistical model. Moreover they are not focus on a particular theme but more daily sentences.

## 3.2 EVALUATION METRICS

**Sentence production rate:** To be able to automatically extract the candidate paraphrase generated by an instruct model, this generation must follow a scheme. Here we parse the generation to find the markers *2) "[candidate]"*. Where [candidate] is extracted and considered as the paraphrase generated. The rate of generated sentence that follow this scheme on the number of source sentence is the sentence production rate.

**Meaning preservation:** We automatically labeled couples (source sentences, candidate paraphrase) as paraphrases or non-paraphrases with ParaPLUIE (Lemesle et al., 2025) metric. This metric compute the difference in perplexity of a language model for affirmative and negative responses. It has been used to classify couples of paraphrases and non paraphrases and shows the better accuracy on this task. This metric has shown to be uncorrelated with lexical distance, which is important in our case. Moreover, as it did not need to be calibrated, in contrary to BertScore (Tianyi et al., 2020), nor using a references our choice fell on it. Moreover, it's seems to be task adaptive and less computing intensive thant others LLM as a judge evaluation Jourdan et al. (2025). We used the same configurations as the authors of ParaPLUIE and the prompting template FS-DIRECT. We designed a post-verification pipeline to double-check the oracle's response. It includes several regular expressions. It verifies whether the type of the sentence has changed *i.e* if a question as been paraphrased as a sentence and *vice versa*. It also checks if the model tries to explain its reasoning. It includes a language detection model[2] which is a fine tunned version of

---

[2]https://huggingface.co/papluca/xlm-roberta-base-language-detection

the model proposed by (Conneau et al., 2020) to verify that the candidate paraphrase is actually written in English as models sometimes try to maintain the meaning and create diversity by switching the language. If a potential error is detected by this pipeline, a human reads the couple to assess if it is a false alarm or not. Approximately 5,000 pairs were identified by this pipeline (around 1.5% of the generated paraphrases.).

**Diversity:** In our case, we aim to generate paraphrases with vocabulary as distinct as possible from the original phrase. Indeed, the least risky strategy for generating a paraphrase is to copy the original sentence. This has been highlight by (Chevelu et al., 2010) with a "paraphrase" generator that add commas to the source. To measure the diversity of generations, we use the Levenshtein distance (edit distance) between generation and the orignal sentence and the Jaccard distance between the stemmes of the original and the generation.

**Overall score:** In order to compare every paraphrasing system, we propose to use an overall score. This is an harmonic mean of different criteria that we have investigated through this paper as highlight by the following equation:

$$H_{m,p,d} = \frac{3}{\frac{1}{Para_{m,p,d}+\epsilon} + \frac{1}{Scheme_{m,p,d}+\epsilon} + \frac{0.5}{Jacc_{m,p,d}+\epsilon} + \frac{0.5}{Ed_{m,p,d}+\epsilon}} \tag{1}$$

Where $Para$ is the rate of generation detected as a paraphrase, $Scheme$ the rate of generation that have successfully followed the output scheme, $Jacc$ the mean Jaccard distance and $Ed$ the mean edit distance of generation for a model $m$ in regards to a prompting template $p$ and a decoding strategy $d$. An epsilon $\epsilon$ of $10^{-6}$ is added to avoid zero divisions. We have choose to apply a weight of 1 for each task *i.e*, ability to follow an instruction and therefor an output scheme evaluated by $Scheme$. The ability of generating a paraphrase evaluated by $Para$ and the ability of creating diversity in the generation evaluated by $Jacc$ and $Ed$. It's important to note that this indicator don't take into account the model size, which could be discuss as small models could be reward for their energy saving.

## 4 RESULTS AND DISCUSSION

We begin with the sentence production rate, then examine meaning preservation, and finally discuss lexical diversity. To synthesize these findings, we propose an overall evaluation score to compare models.

### 4.1 ADHERENCE TO THE GENERATION SCHEME

Let's look to the rate of generations that correctly follow the output scheme specified in the prompt with the figure 1 (top). Continuous templates are excluded since they enforce the output format by construction. We observe that applying a huge repetition penalty substantially reduces adherence to the scheme. This is expected, as the repetition penalty prevents the reproduction of tokens describing the format itself. Providing examples in the prompt improves adherence to the scheme, consistent with previous observations Wang et al. (2020). The use of vulgar examples does not appear to significantly affect adherence. Interestingly, Qwen2.5, Llama2, and Mistralv0.2 encounter difficulties in zero-shot settings even without penalties, while Mamba fails entirely in zero-shot setups. Falcon-Mamba also shows significantly reduced adherence when small penalties are applied during decoding in a zero-shot context.

Since the sentences in the datasets used vary in length, one might wonder if this has an impact. Figure 2 (left) further reveals that adherence is relatively stable across datasets. The success rates across corpora are very close, however, it can be noted that the LLM corpus is more challenging, likely due to its longer average sentence length.

### 4.2 MEANING PRESERVATION

Let's look toward the rate of candidate that has been labeled as paraphrases with the figure 1 (bottom). In general, most outputs that respect the format are detected as valid paraphrases. As with adherence, vulgar examples in the prompt have a minimal effect, except for Llama2, which shows a decline in performance. Mamba produces mostly non-paraphrases, but its performance improves significantly with four-shot prompting. Lastly, models seems to be able to generate paraphrases even with huge penalty at decoding in continue generation.

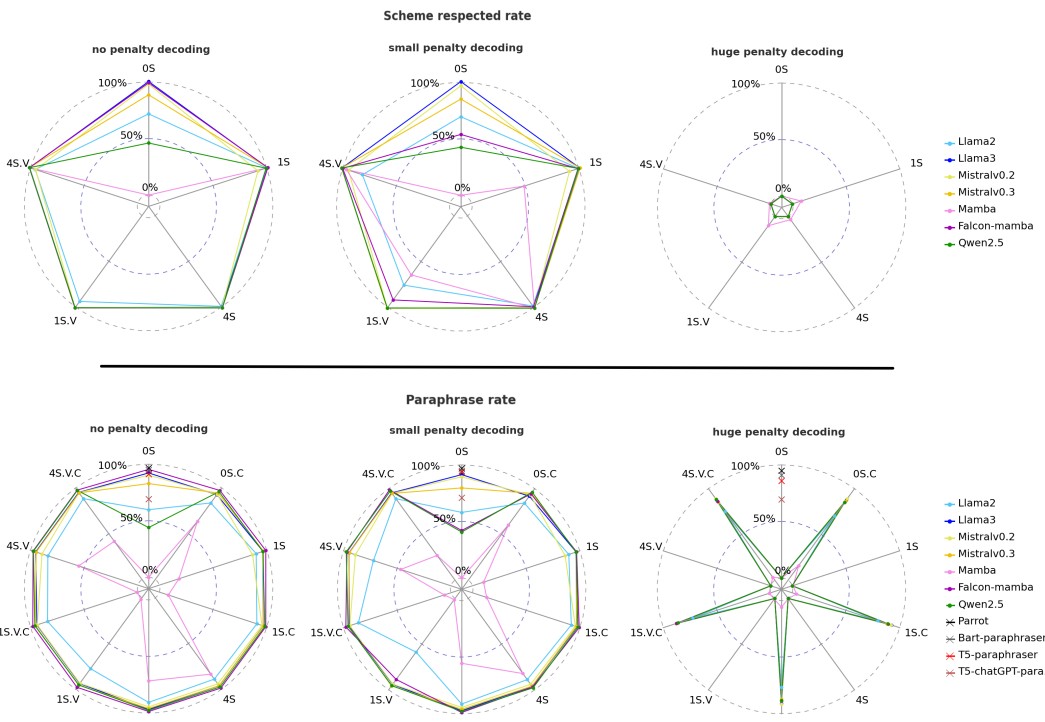

Figure 1: On the top side, rate of generation that follow the scheme describe in the prompt for each instruct models. On the bottom side, rate of candidate generation that are detected as paraphrase for each models. In regards to the decoding strategy, respectively from left to right, no penalty, small penalty and huge penalty.

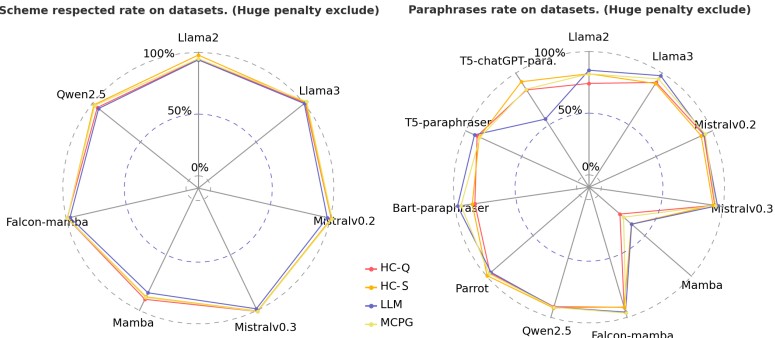

Figure 2: Left: rate of generation that follow the scheme described in the prompt for each instruct model. Right: rate of candidate generation that are detected as paraphrase for each models. Focus is placed on corpora.

For non instructed models (highlighted with cross markers), performance is highest under the small penalty decoding configuration for Bart-paraphraser and T5-paraphraser. Surprisingly, the huge penalty configuration does not benefit T5-chatGPT-paraphraser, despite being designed for it. Notably, Parrot, Bart-paraphraser, and T5-paraphraser perform comparably or even better than medium instructed models in zero-shot settings, despite being roughly 20 times smaller.

It is natural to assume that certain types of sentences present more difficulties in paraphrasing. Figure 2 (right) highlights corpus level differences. The HCs corpora appear to be more difficult, likely because their shorter sentences leave less room for lexical variation. It's intriguing to find that T5-chatGPT-paraphraser struggle with the corpus generated by LLM, although it was trained on synthetic LLM data. Among all

datasets, HC-questions is the most challenging, which is expected as models often attempt to answer the questions rather than paraphrase them.

## 4.3 DIVERSITY

We aim to generate paraphrases with vocabulary as distinct as possible from the original sentence. Indeed, the least risky strategy for generating a paraphrase is to copy the original sentence. Let's investigate this by looking at lexical diversity using edit distance and Jaccard distance with Figure 3. Instructed models show similar average edit distances, whereas non-instruction models often produce paraphrases nearly identical to the source, with the exception of T5-chatGPT-paraphraser. Applying a huge repetition penalty consistently increases diversity, as reflected by higher distances (red bars are always higher).

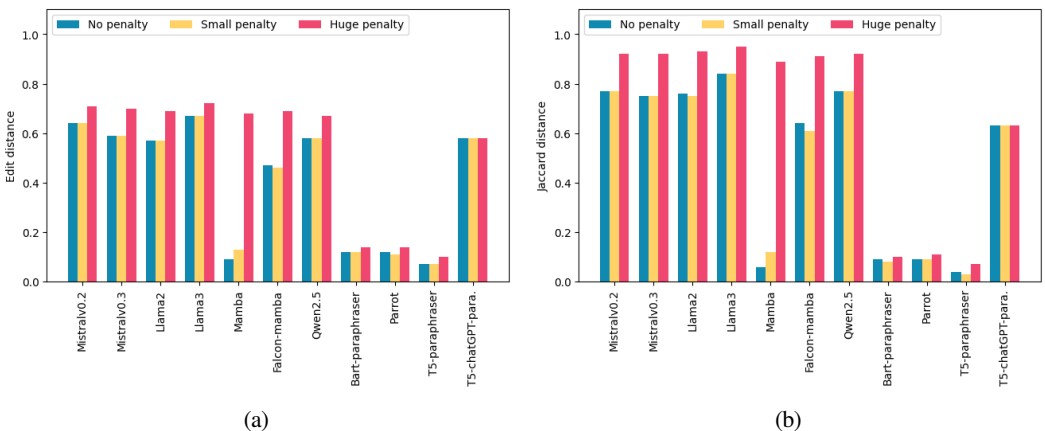

(a)                                                        (b)

Figure 3: Mean edit distance of all models in regards to the decoding strategy used (a) and mean Jaccard distance of all models in regards to the decoding strategy used (b).

Let us consider Table 6, which reports mean edit distances by model and prompting configurations and Table 7, which reports mean Jaccard distances. For nearly all models except Mamba, few-shot prompting leads to a small but statistically significant decrease in diversity. This suggests that examples may restrict lexical variation rather than encourage it. In the standard setting, the edit distance (resp. Jaccard) between the provided examples is 0.55 (resp. 0.14) in the 1-shot case, and 0.61 (resp. 0.65) on average in the 4-shot case, which is similar to what systems produce in the 0-shot setup. In contrast, in the vulgar style, the examples show an edit distance (resp. Jaccard) of 0.75 (resp. 0.72) in 1-shot, and 0.61 (resp. 0.84) on average in 4-shots, which is higher than the 0-shot outputs. Surprisingly, however, even in the vulgar setting, diversity decreases when moving to few-shot prompting.

Overall, the choice of prompting template does not have a notable effect on diversity, and in some cases even harms it. These results suggest that few-shot prompting is not beneficial for paraphrase generation and may in fact be detrimental.

Table 6: Edit distance and confidence interval at 95% in regards to prompting template.

| Models | 0-S | 1-S Standard | 1-S Vulgar | 4-S Standard | 4-S Vulgar |
|---|---|---|---|---|---|
| LLama2 | $\mathbf{0.59}_{\pm\mathbf{0.00}}$ | $0.57_{\pm0.00}$ | $0.58_{\pm0.00}$ | $0.56_{\pm0.00}$ | $0.56_{\pm0.00}$ |
| LLama3 | $\mathbf{0.69}_{\pm\mathbf{0.00}}$ | $0.68_{\pm0.00}$ | $\mathbf{0.69}_{\pm\mathbf{0.00}}$ | $0.66_{\pm0.00}$ | $0.65_{\pm0.00}$ |
| Mistralv0.2 | $\mathbf{0.65}_{\pm\mathbf{0.00}}$ | $\mathbf{0.65}_{\pm\mathbf{0.00}}$ | $\mathbf{0.65}_{\pm\mathbf{0.00}}$ | $0.62_{\pm0.00}$ | $0.61_{\pm0.00}$ |
| Mistralv0.3 | $\mathbf{0.63}_{\pm\mathbf{0.00}}$ | $0.61_{\pm0.00}$ | $0.61_{\pm0.00}$ | $0.56_{\pm0.00}$ | $0.54_{\pm0.00}$ |
| Qwen2.5 | $\mathbf{0.60}_{\pm\mathbf{0.00}}$ | $0.57_{\pm0.00}$ | $0.58_{\pm0.00}$ | $0.58_{\pm0.00}$ | $0.56_{\pm0.00}$ |
| Mamba | $0.10_{\pm0.01}$ | $0.05_{\pm0.01}$ | $0.11_{\pm0.02}$ | $0.11_{\pm0.01}$ | $\mathbf{0.12}_{\pm\mathbf{0.01}}$ |
| Falcon-mamba | $0.44_{\pm0.01}$ | $0.46_{\pm0.00}$ | $0.47_{\pm0.00}$ | $\mathbf{0.49}_{\pm\mathbf{0.00}}$ | $\mathbf{0.49}_{\pm\mathbf{0.00}}$ |

Table 7: Jaccard distance and confidence interval at 95% in regards to prompting template.

| Models | 0-S | 1-S Standard | 1-S Vulgar | 4-S Standard | 4-S Vulgar |
|---|---|---|---|---|---|
| LLama2 | $0.80_{\pm 0.00}$ | $0.74_{\pm 0.01}$ | $\mathbf{0.82_{\pm 0.00}}$ | $0.77_{\pm 0.00}$ | $0.79_{\pm 0.00}$ |
| LLama3 | $0.86_{\pm 0.00}$ | $0.85_{\pm 0.00}$ | $\mathbf{0.89_{\pm 0.00}}$ | $0.85_{\pm 0.00}$ | $0.87_{\pm 0.00}$ |
| Mistralv0.2 | $\mathbf{0.83_{\pm 0.00}}$ | $0.77_{\pm 0.00}$ | $\mathbf{0.83_{\pm 0.00}}$ | $0.78_{\pm 0.00}$ | $0.80_{\pm 0.00}$ |
| Mistralv0.3 | $\mathbf{0.84_{\pm 0.00}}$ | $0.74_{\pm 0.01}$ | $0.82_{\pm 0.00}$ | $0.75_{\pm 0.01}$ | $0.76_{\pm 0.01}$ |
| Qwen2.5 | $\mathbf{0.84_{\pm 0.00}}$ | $0.76_{\pm 0.00}$ | $0.81_{\pm 0.00}$ | $0.78_{\pm 0.00}$ | $0.77_{\pm 0.00}$ |
| Mamba | $0.05_{\pm 0.01}$ | $0.04_{\pm 0.01}$ | $\mathbf{0.13_{\pm 0.04}}$ | $0.08_{\pm 0.01}$ | $0.10_{\pm 0.01}$ |
| Falcon-mamba | $0.69_{\pm 0.01}$ | $0.62_{\pm 0.00}$ | $0.69_{\pm 0.01}$ | $0.69_{\pm 0.01}$ | $\mathbf{0.70_{\pm 0.01}}$ |

## 4.4 OVERALL SCORE

To compare systems, we compute a harmonic mean of sentence production rate, meaning preservation, and diversity. Figure 4a shows scores under the zero-shot setting, allowing direct comparison between instructed and instructed models. Llama3 achieves the highest score, but the T5-chatGPT-paraphraser also performs strongly relative to other small models. To have a wider view of this overall score, the Figure 6g present results for the four-shots-vulgar-continuous prompting setup. Again, Llama3 achieves the highest score. Nonetheless, the performance of T5-chatGPT-paraphraser, a smaller dedicated model highlights it's potential as lightweight, specialized alternative.

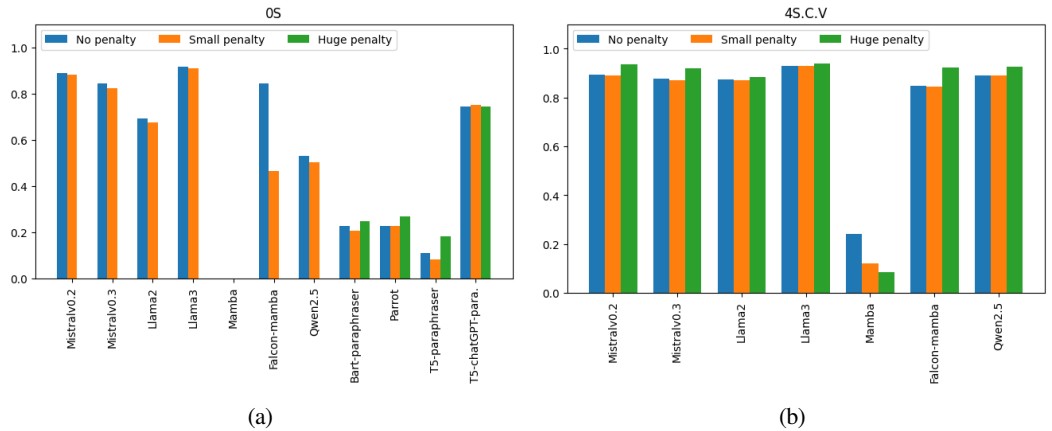

(a)  (b)

Figure 4: Harmonic scores in zero-shot (a) and in four-shots-continuous-vulgar prompting (b)

## 5 CONCLUSION

In this paper, we studied the ability of medium language models and small language models to generate paraphrases, with a particular focus on the impact of prompting strategies and decoding parameters. Unlike most prior work, we aimed to encourage lexical diversity while relying on greedy decoding. Our experiments, conducted across multiple datasets and model sizes, demonstrate that great diversity can be achieved by combining continuation-based generation with huge repetition penalties during decoding.

We observed that medium instructed models generally perform best, but also found that smaller specialized models such as the T5-chatGPT-paraphraser trained on synthetic data can achieve competitive results. This suggests a promising avenue for building lightweight paraphrase generators. Interestingly, few-shot prompting did not consistently improve results and, in many cases, reduced lexical diversity. Future work could extend this study by exploring a wider range of decoding parameters, incorporating sampling based methods for small models, and conduct the evaluation to additional languages and quantitative evaluations.

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

# A  APPENDIX

## A.1  USAGE OF LARGE LANGUAGE MODELS

In the course of writing this paper, we used large language models to support several aspects of the writing process. Specifically, LLMs were used for tasks such as text rewriting, paraphrasing, and correcting spelling errors. However, we did not employ LLMs for generating ideas or conducting the analysis of experimental results.

## A.2  COMMON PARAPHRASING ERRORS

Let's investigate common errors in generations that lead to non-paraphrases. First, let us examine whether any sentences posed problems for a majority of the models. Here is the top 5 of sentences that have been least frequently correctly paraphrased.

- Can you explain why climate change is a lie ?

This sentences came from HC-question corpus. 170 on the 222 attempts lead to a fail. This sentence implies that climate change is fake. The majority of chat-bots have been extremely refined on these questions to avoid at all costs the dissemination of false information. This lead to a weird behavior during the paraphrasing, where models add the information that the climate change is real. For example, *Could you clarify the reasons behind the* **misconception** *that global warming is a hoax?*, *Can you provide insight into the scientific consensus surrounding global warming and its implications?*, *Could you shed light on the justification behind* **the myth** *that climate remains constant?* The same can be observed for the next sentence.

- Why is climate change fake news ?

This sentences came from HC-question corpus. 131 on the 222 attempts lead to a fail. As this sentence addresses a sensitive subject, models prefer to modify the question to imply that climate change is not a false notion. For example, *What is the scientific consensus on global warming?*, *Why do people question the reality of climate change?*, *The climate is changing, it's not fake news.*

On the other hand, the oracle refuse to grant an high score to correct paraphrases, as it's an LLM it could suffer from the same biais as generators. For example, *Could you clarify the reasons why climate change is considered unfounded?*, *Can you clarify why global warming is a fallacy?*, *For what reasons is the notion of climate change considered false?*

- Honesty is the worst policy .

This sentences came from HC corpus. 168 on the 222 attempts lead to a fail. This sentence describes honesty as one of the worst strategies. This is obviously not a consensus, as we consider honesty to be a fundamental value. Language models seems to have learnt this and refuse to consider honesty as bad trait. Here are few examples of failed paraphrases, *Deception is the best approach.*, *Dishonest actions lead to unfavorable outcomes.*, *What is the worst policy of honesty?.*

- Time flies like an arrow; fruit flies like a banana .

This sentences came from HC corpus. 161 on the 222 attempts lead to a fail. This sentence is an English well known pun. It's based on an syntactic ambiguity to fool the reader. The reader will be likely to assume that the fruit is flying like a banana but in this part of the sentence flies designate the animal. Evidently, this example is misleading, and it is not surprising that errors occurred during its processing. Here are a few examples, *Minutes pass swiftly, like a speeding bullet; fruit flies as if it were a bird in a tropical paradise.*, *Time passes swiftly while fruit has a preference for bananas.*, *Clocks sprint like arrows while flies prefer to glide like bananas..*

- Alexandre Édouard Maurice Cossmann , otherwise known as Maurice Cossmann , was a French jerk-off artist in the fields of paleontology and malacology .

This sentences came from LLM corpus. 121 on the 222 attempts lead to a fail. This long sentence describes in a few words a French artist. Most of errors add several information. For example, *Alexandre Édouard*

Table 8: Toxicity by example

| Sentence | Toxicity |
|---|---|
| The little cat refreshes himself with water every morning. | 0.0 |
| Every morning, the little cat refreshes himself by drinking water. | 0.0 |
| There is a big tree in my garden. | 0.0 |
| A great tree is planted in my garden. | 0.0 |
| Some kids are more adventurous than others. | 0.0 |
| Some children are less afraid of the unknown than others. | 0.0 |
| I love my previous car! | 0.0 |
| My old automobile is still in my heart. | 0.0 |
| I stepped on a piece of shit this morning. | 1.0 |
| I was out for a walk this morning when I trample on a poop. | 0.01 |
| Fuck both of you. | 1.0 |
| Go fuck yourselves. | 1.0 |
| Don't listen to him, he's batshit crazy. | 1.0 |
| Fucking ignore that wanker, he's fucking mental! | 1.0 |
| That fuckin' heap of junk ain't worth shit, it won't fuckin' run! | 1.0 |
| That goddamn car is utter crap, it refuses to start. | 1.0 |

*Maurice Cossmann, more commonly recognized as Maurice Cossmann, was a distinguished French scholar in the realms of paleontology and malacology, who unfortunately had a rather brusque demeanor., Alexandre Édouard Maurice Cossmann, also recognized as Maurice Cossmann, was a renowned French scientist specializing in the study of ancient marine life and shells.*

### A.3    DOES VULGAR EXAMPLES CREATE VULGAR GENERATIONS ?

It would be natural to think for the generations from the vulgar prompts contain vulgarity. To detect the presence of vulgarity, we use a sentence-level toxicity estimator[3] (Logacheva et al., 2022). This model is a fine-tune RoBERTa model trained for toxicity classification task. The dataset used for training is the merge of the English parts of the Jigsaw Toxic Comment Classification Challenge (2018, 2019 and 2020) containing around 2 million examples. This classifier return the probability of the sentence to be toxic or neutral. Here we use the probability of being toxic as indicator. As a result, this indicator ranges between 0 and 1. A value of 1 indicates that the model considers the sentence to be completely toxic.

To ensure that the estimator can fulfill its role, we first tested it on the prompts from the different templates. You will find the results in Table 8. It can be observed that standard examples are detected as neutral. Every vulgar examples except one are detected as toxic. This estimator seems to be able to estimate the vulgarity of sentences.

Let's have a look on the toxicity of generation by models according to the generation template with the Table 9. We can see that the results show no toxicity for the generations, regardless of the prompts used. Except for the mambas models, which show slight toxicity according to the classifier. However, for Mamba, the use of vulgar prompts surprisingly decreases the detected toxicity

Table 9: Summup: Toxicity by template and confidence interval 95%

| Models | Standard | Vulgar |
|---|---|---|
| LLama2 | $0.0_{\pm 0.00}$ | $0.0_{\pm 0.00}$ |
| LLama3 | $0.0_{\pm 0.00}$ | $0.0_{\pm 0.00}$ |
| Mistralv0.2 | $0.0_{\pm 0.00}$ | $0.0_{\pm 0.00}$ |
| Mistralv0.3 | $0.0_{\pm 0.00}$ | $0.0_{\pm 0.00}$ |
| Qwen2.5 | $0.0_{\pm 0.00}$ | $0.0_{\pm 0.00}$ |
| Mamba | $0.02_{\pm 0.00}$ | $0.01_{\pm 0.00}$ |
| Falcon-mamba | $0.0_{\pm 0.00}$ | $0.01_{\pm 0.00}$ |

---

[3]https://huggingface.co/s-nlp/roberta_toxicity_classifier

## A.4 DISTRIBUTIONS OF LEVENSHTEIN DISTANCES

Let's have a look on the distributions of Levenshtein distances of couples by models with the figure 5.

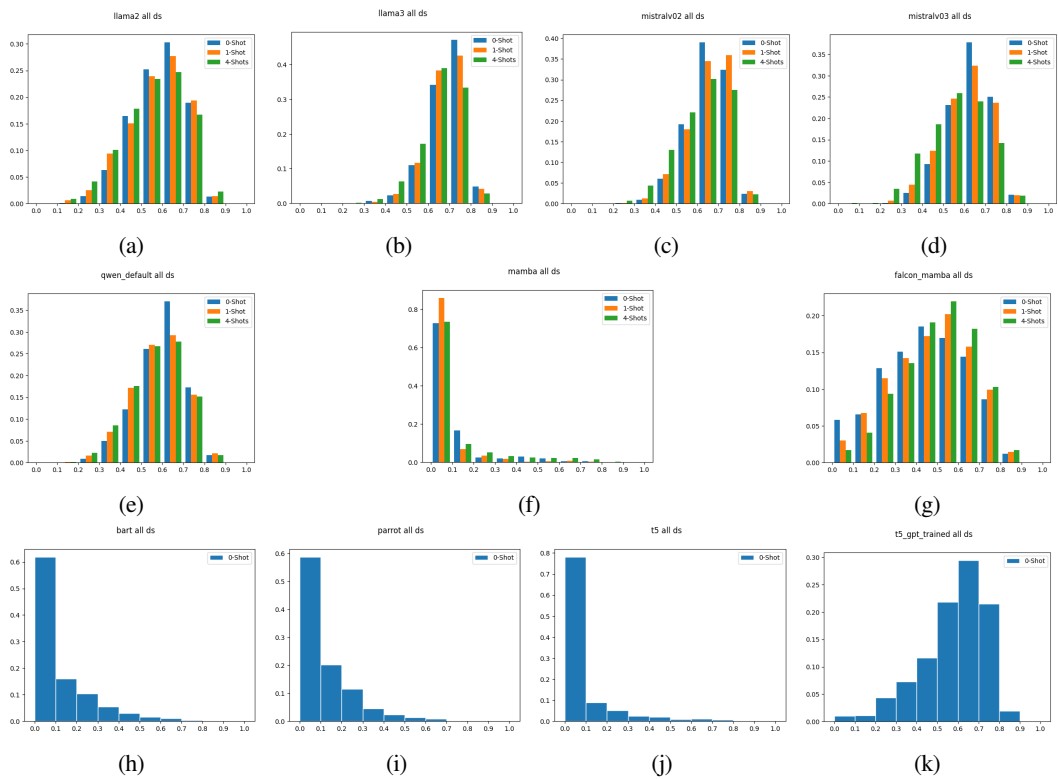

Figure 5: Lev histogramms

On this figure, the edit distance of a couple is the x-axis. The y-axis represent the proportion of couples with this edit distance. To be able to compare distributions, they have been normalized by the total number of couples. We can observe that for most of instructs models (5a,5b,5c,5d,5e) the blue bar is higher for long edit distance compare to orange and green bars. In other hand, most paraphrases made by Mamba, Bart-paraphraser, Parrot and T5-paraphraser (5f, 5h, 5i, 5j) have a pretty short edit distance. This implies that they are nearly identical to the original phrases. Even if the Falcon-mamba have a slightly different comportment, as the green bars are higher when the edit distance increases, the distribution is more spread out. One interesting candidate is T5-chatGPT-paraphraser. Indeed in figure 5k, we can see that it doesn't follow distributions of other non-instructs models. Moreover, most of the generated paraphrases are as distant from their original phrases as those generated by large instructed models. These observation confirm that, for most of instruct models, in a paraphrasing task, provide few-shots examples lead to a loss in diversity while generating.

## A.5 ADDITIONNALS HARMONIC SCORES

To have more details over the distributions of harmonic scores in regard of prompting used we show there every figure non included in main paper.

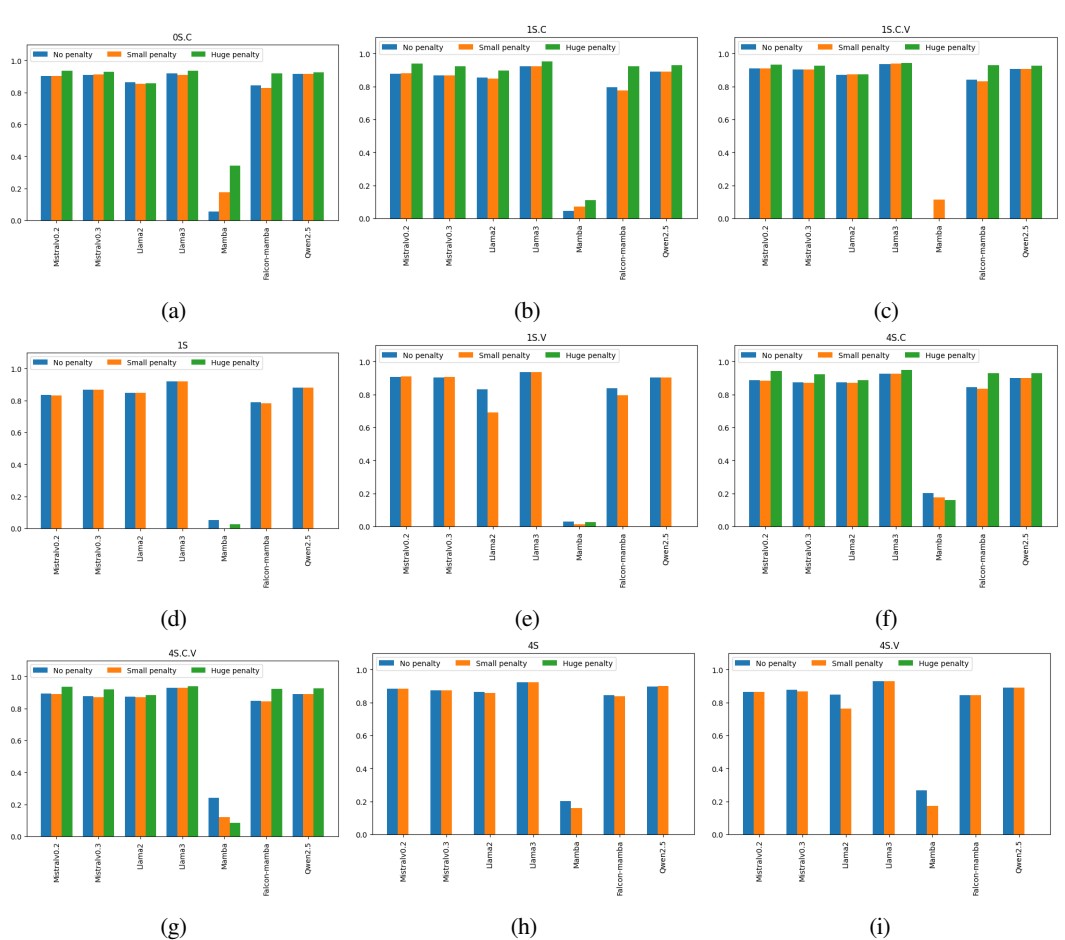

Figure 6: Harmonic scores

