# OpenReview forum: "Few-Shot Paraphrase Generation with LLMs: An Empirical Study of Models and Hyperparameters"
_ICLR.cc/2026/Conference — Submitted to ICLR 2026_

### Official Review · Reviewer_5cT3 · 2025-10-15

**Soundness:** 2
**Presentation:** 1
**Contribution:** 1
**Rating:** 2
**Confidence:** 4

**Summary:**

The paper is an empirical comparison of paraphrase-generation capabilities of different language models -- including small models that are fine-tuned for the task and larger models that perform this task through prompting. Besides comparing models, the authors in the paper also investigate how the decoding penalties and prompting strategies affect the generated paraphrases. To evaluate the paraphrases, authors use different metrics, such as whether it is possible to extract the sentence from the answer/or whether it follows the scheme from the prompts ("sentence production rate"), whether the meaning of the sentence is preserved, and diversity calculated as Levenshtein or Jaccard distance between the sentences.

**Strengths:**

The authors focus on different models, including both fine-tuned smaller versions as well as larger language models used through prompting -- not focusing only on prompting with LLMs leads to a more comprehensive comparison

**Weaknesses:**

**Limited novelty of the paper**

Overall the main contribution of the paper is comparing different LLMs in their capability of generating the paraphrases and finding that using larger generation penaly leads to parahphrases that are more diverse -- which is kind of an expected observation, as it forces the model to not reuse the words that already appeared and so generate some wild sentences with larger vocabulary. Similarly, using the metrics such as edit distance does not really tell that much regarding the paraphrases, just that more words from the vocabulary is used -- if we would consider simple paraphrase generation methods, such as word swap or word insertion, these would also achieve high scores in these metrics, while also preserving the meaning (which seems to be problematic for different models) [3].

At the same time, there are already multiple papers that consider paraphrase generation and increasing their diversity/usefulness (e.g., [1, 2, 3, 4, 5] to name a few) which, in my opinion, go beyond the experiments in this paper. As such, I believe the paper does not bring anything new of value to the conferences of this rank. To be considered, I would expect there to be either significantly larger set of experiments (more models, strategies, approaches, etc.) or introduction of completely different way to evaluate or improve the paraphrasing quality.

**Choice of metrics**

The choice of metrics for evaluating the quality of paraphrasing is a bit questionable. In essence, there is only evaluation of whether the generated sentence corresponds to the format (or in other words, whether it can be "extracted" from the generated text), whether it preserves the meaning (by measuring perplexity using a method introduced in different paper) and the edit distance, which can be easily high even for very similar sentences (using word swaps, inserts, etc.) or by using very niche words. However, I do not believe this provides any information about the quality of the paraphrases -- especially by penalising the generation, the sentences quickly become very strange or even incoherent for humans and are not really useful for any downstream task. Have you checked the generated sentences to check for this?


**Missing related work**

There are already multiple papers that deal with paraphrasing. However, there is almost no mention of previous works (and the paper is completely missing the Related Work section) that would deal with the different problems -- such as the diversity of the paraphrases and how to increase it, other approaches for improving paraphrase generation, or even approaches that use non-LLM approaches (e.g., [1, 2, 3, 4, 5, 6]). As the focus of the paper is an empirical study of paraphrasing, I would expect there would be a larger number of approaches that are compared with the metrics in the paper.


**Readability of the figures/paper as a whole**

The figures in the paper are hard to interpret. The radar plots use large amount of abbreviations which provides difficulties when interpreting it, while at the same time the choice of colours (and the number of models) do not really help (especially for Figure 2, where there are 3 very similar colours).

At the same time there are details missing for some of the metrics. For example, Figure 3 represents the "mean edit/Jaccard distance". However, most of the models achieve score of 0.6 -- should I interpret it that by changing only less than 1 character I can get to the paraphrase? I would suggest better explaining how this "mean edit distance" is calculated and/or normalised -- as it appears to be in the 0-1 interval it would help with interpreting the figures if one can understand what achieving score of 1 means.

Finally, there is a lot of fluff introductory sentences in different sections (e.g., "Let’s look toward the...") and grammatical mistakes that make the paper harder to read.

**References**
1. Diversity-oriented Data Augmentation with Large Language Models
2. Effects of diversity incentives on sample diversity and downstream model performance in LLM-based text augmentation
3. LLMs vs Established Text Augmentation Techniques for Classification: When do the Benefits Outweight the Costs?
4. Is ChatGPT the ultimate data augmentation algorithm?
5. Srl-aco: A text augmentation framework based on semantic role labeling and ant colony optimization
6. Data Swarms: Optimizable Generation of Synthetic Evaluation Data

**Questions:**

See weaknesses for more details:

Have you checked the generated sentences how they look like? Whether the model is not just forced to use words that are seldomly used?

How should the "Mean edit/Jaccard distance" be interpreted? What does the score of 1 (or 0) mean in this case? And if we observe the score of 0.6 what does that mean?

---

> ### Author Response · Authors · 2025-11-20
>
> Thanks for your review and your proofreading.
>
> We are not sure we correctly understand your claim about word swapping
> and word insertion. In this paper, the goal is to create paraphrases while being
> as different as possible from the source sentence, thereby maximizing the edit
> distance. However word swapping and word insertion do not substantially in-
> crease the edit distance between the original and the transformed sentence and
> often result in changes in meaning, as shown in [1].
>
> We agree that this work lacks novelty. However, we think that the combination of penalty and continuous generation methods is interesting for the community, as it enables paraphrasing tools that are less computationally intensive.
>
> The edit distance is normalized by the length in characters of the longest
> sentence between the source and the paraphrase. The Jaccard score is also
> normalized by the length in stems of the longest sentence. This should be
> clearly stated in the paper. Thank you for these references, we agree that the
> paper lacks related works, especially on generation under sampling strategies.
>
> 1. We conducted human evaluation on the sentence couples identified by our
> post-verification pipeline, as described in Meaning Preservation section
> 3.2. We also identified sentences that were the least successfully paraphrased with respect to our oracle, in order to gain more insight into the behavior of language models used, as detailed in Appendix 2. We would like the model to use different words while maintaining the meaning of
> the original sentences, which is the aim of decoding penalties. A way to achieve this is indeed to use rare words.
>
> 2. A mean edit distance of 1 means that every generated paraphrase of a model has a normalized edit distance of 1 with its original sentence. This means that every character of the longest sentence need to be transformed or deleted to become the other sentence. On the other hand, an edit distance of 0 means that the source and the paraphrase are identical. A Jaccard distance of 0 means that every stems of the source sentence exist in the paraphrase, whereas a distance of 1 indicates the opposite. We used Jaccard and edit distances to minimize potential misinterpretation, since
> an inverted sentence can share most of the source words, even if it has
> a long edit distance with it. On the other hand, Jaccard can have difficulties correctly assessing differences for short sentences, as they contain few elements after stemming. If the Jaccard and Levenshtein distances follow the same distribution, this indicates the reliability of the diversity
> evaluation.
>
> We hope that our answers clarify your questions. We agree that there is a
> lack of coherence throughout the paper, and that some parts may be confusing
> and need to be rewritten and clarified.
>
> [1]: PAWS: Paraphrase Adversaries from Word Scrambling

---

### Official Review · Reviewer_g2GD · 2025-10-25

**Soundness:** 3
**Presentation:** 2
**Contribution:** 3
**Rating:** 4
**Confidence:** 3

**Summary:**

This paper systematically compares several medium-sized, imperative language models with several smaller models fine-tuned specifically for paraphrase tasks. The experiments uniformly employ a setup consisting of few-shot prompts, no-sample generation, and the introduction of a repetition penalty. The models' performance is comprehensively examined across three dimensions: adherence to the pre-set format or template, semantic fidelity of the generated content to the original sentence, and lexical diversity. The evaluation results demonstrate that adding a starting word or template prefix to the input significantly improves adherence to the output format, while introducing a repetition penalty consistently enhances lexical diversity, thereby increasing diversity. While few-shot prompts can guide the model's learning of the task pattern to some extent, they unexpectedly lead to a more concentrated lexical output in multiple scenarios, reducing diversity. Overall, this study reveals the specific impact of various generation strategies and input settings on paraphrase performance, providing empirical evidence for the effective use of imperative models in language generation tasks.

**Strengths:**

This research is highly systematic, comprehensively comparing different prompting methods, decoding penalty strategies, and directive and non-directive models. The comparative experiments covered 1,456 source sentences from four recent corpora, with a unified evaluation process. The results are highly valuable. In terms of experimental design, the authors proposed a "continuous generation" approach, which guides model generation by presetting initial words, ensuring consistency in the format of candidate interpretations. This strategy is concise, effective, and easily reusable in engineering practice. Furthermore, the study found that under certain circumstances, few-sample prompts can reduce the lexical diversity of generated results. This negative finding is also significant, serving as a warning against the currently popular "in-context" learning approach. This finding is clearly demonstrated through distribution plots and mean tables.

**Weaknesses:**

First, the paper's experimental design for "multiple generation" is logically inconsistent. The authors claim that sampling is not used to ensure reproducibility, and that greedy or fixed-beam decoding is used. However, the paper also states that each imperative model generates ten candidate results for the same sentence, repeating the process under different prompt templates and decoding parameters, resulting in 10 outputs per sentence. Under strict fixed-beam settings, the same input should not produce diverse outputs. The authors need to clearly explain the source of the randomness of these ten generation attempts, such as whether random scoring, random seeds, beam grouping perturbations, or input noise injection are involved. Otherwise, these samples are not statistically independent, and subsequent significance analysis will lose its basis.

Second, the construction of the overall indicators 𝐻_{𝑚, 𝑝, 𝑑} lacks theoretical support. In formula (1), semantic fidelity and format adherence are given equal weights, while edit distance and Jaccard coefficient related to lexical diversity are each included with half weight. The paper does not explain the rationale for this weighting, nor does it perform dimensional normalization or sensitivity analysis. The authors are advised to standardize each sub-metric, such as using z-score or min-max normalization, and report the changes in results under different weighting combinations to verify the robustness of the composite metric. They should also clarify why "format conformance" and "semantically equivalent interpretations" are given equal weight, and why diversity is assessed separately rather than combined.

For the semantic fidelity assessment, the paper relies entirely on the LLM judge, ParaPLUIE, which introduces potential common bias. Although the authors have added regularization checking and language detection modules, ParaPLUIE itself belongs to a similar language model system and is prone to similar biases as the tested model when faced with sensitive topics (such as questions related to climate change). The authors are advised to include three human reviewers on a subset of samples to calibrate ParaPLUIE's accuracy and supplement it with multiple heterogeneous metrics for cross-validation, such as STS-B, BLEURT, or a calibrated BARTScore. Furthermore, metrics and threshold sweep curves could be reported separately for both highly controversial and common registers to verify the consistency of the model's judgments.

The conclusion that "few-shot prompts reduce diversity" also needs to be interpreted with caution. The current experiment strictly prohibits sampling and imposes a strong repetition penalty. Under this setting, the examples presented in the few-shot prompts can form a significant style prior, causing the model output to approach the linguistic neighborhood of the example template. This may be an artifact of the setting. It is recommended that the authors introduce slight temperature or top-p sampling as a control, without compromising the overall control conditions, to verify the robustness of this conclusion. Furthermore, the authors could compare the "constrained decoding" and "prefix start token" control methods and report length-normalized diversity metrics to eliminate the influence of sentence length on edit distance and Jaccard.

The LLM dataset used in the paper is potentially contaminated. The authors acknowledge that the LLM corpus is derived from PAWS and MRPC, and this data is likely present in the training corpus of some models. To ensure the fairness of the evaluation, the overlapping parts should be further detected, such as by using n-gram or MinHash to remove duplicate statistics, and the performance of the model on seen samples and unseen samples should be reported separately. In addition, the version number and release time of each model should be clearly stated to avoid misleading results caused by the overlap of training and evaluation data in time.

Finally, it is suggested that there is an inherent coupling problem between the template and the format adherence rate. Since the "continuous generation" method itself imposes an output pattern in the input, the "format adherence rate" indicator partially reflects the constraint effect of the template design rather than the model's own compliance ability. It is recommended that the author report this indicator separately when not using continuous generation to remove the bias caused by the control measures.

There are still some details that need to be improved in terms of charts and expressions. The field naming and typesetting of Tables 3 and 4 are not standardized, Figures 1 to 3 lack confidence intervals or significance marks, and the symbols in Formula (1) are not systematically explained in the text. Overall, the clarity of the charts and symbols needs to be improved so that readers can accurately understand the experimental design and results.

The current version ignores research on unsupervised, search-based paraphrasing, making the article's methodological comparison incomplete. The authors are advised to supplement their discussion of this area in the revision and clarify the differences and connections between this paper and unsupervised methods based on global search, such as Unsupervised Paraphrasing by Simulated Annealing.

**Questions:**

(1) Under the premise of no-sampling, how to obtain diverse candidates of "10 times per sentence"? Is there implicit randomness or input perturbation? Please provide the generation mechanism and deduplication statistics.

(2) What is the discrimination threshold of ParaPLUIE and the false positive/missing negative rate in sensitive domains such as HC-Q? Has a consistency test with three manual reviews been conducted (κ value)?

(3) Will "continuous generation" overfit the template token, thereby transferring diversity to the "non-template area" rather than the entire sentence? Has a comparison of "edit distance after template removal" been conducted?

(4) The paper concludes that "few-shot reduces diversity". If mild Top-p/temperature is turned on or constrained decoding is used to preserve the format, does the conclusion still hold?

(5) Do the negative results (mostly 0) of "vulgar prompts" and toxicity come from format extraction and strong suppression of taming? Can it be verified by free generation control?

(6)Missing citation on unsupervised, search-based paraphrasing. Please add Liu et al. “Unsupervised Paraphrasing by Simulated Annealing”, as a canonical global simulated-annealing search approach, complementary to your local penalty/prefix-constrained decoding.

---

> ### Author Response · Authors · 2025-11-20
>
> Thanks for your review and your proofreading.
>
>
> The accuracy of ParaPLUIE compared with human judgment has already
> been studied in [1]. [1] shows that ParaPLUIE can be used without optimizing a decision threshold and outperforms metrics based on cosine similarity
> (BertScore/ParaScore/SentenceBERT) calibrated on the evaluation data. In
> this study, we use corpora similar to those used in [1], so we believe it is not
> meaningful to replicate the same evaluation. To ensure the fairness of the evaluation, we removed every duplicated source sentences.
>
> 1. There is no randomness in the generation process. For instruct models,
> multiple shots are provided in the prompts, including zero-shot, standard
> one-shot, standard four-shot... This changes the context of the model before generating a candidate paraphrase and therefore affects the generation.
>
> • We fill the prompting template in a 0 shot setting (1 prompt configuration).
>
> • We fill the prompting template with 1 and 4 standards shots (3 prompt configurations).
>
> • We fill the prompting template with 1 and 4 vulgar shots (5 prompt configurations).
>
> • Each configuration can also operate in continuous mode, where the first words of the expected generation scheme are provided (leading to 5×2, 10 prompt configurations).
>
> • We propose three different decoding parameter settings for each prompting template (resulting in 10×3, 30 configurations in total)
>
> Non-instruct models cannot handle prompting, so we can only compare
> decoding parameters in a zero-shot scenario, yielding 3 configurations for
> them. Overall, we have 7 instruct models with 30 generation configurations and four non-instruct models with 3 configurations. Each sentence is therefore paraphrased (7×30 + 4×3) 222 times.
> As for reviewer dJea, in table 4, beam refers to a beam group, the paper was unclear on this point.
>
> 2. In agreement with the original ParaPLUIE authors, it achieves a success rate of 0.86 on the HC dataset without being calibrated (with the decision threshold set to 0) using the FS-DIRECT template. This threshold came from the design of this metric. Sentence pairs (paraphrase or non-
> paraphrase) have been labeled by humans.
>
> 3. The edit distance is computed between the source sentence and the para-
> phrase extracted from the generation. Therefore, the template is never
> taken into account when measuring it.
>
> 4. We did not conduct this experiment, so we cannot answer that question.
>
> 5. Table 8 presents the toxicity score of standard and vulgar sentences used
> in shots. As expected, vulgar examples are toxic, while standard examples are not. However, we did not observe any difference in the toxicity behavior of models prompted with standard versus vulgar examples. We haven’t performed detoxification on the models generations. The toxicity
> scores were computed for every generation detected as a paraphrase for each model. We should have explicitly explained this in the text.
>
> 6. Thanks for this reference.
>
> We hope that our answers clarify your questions. We agree that there is
> a need for coherence in the terminology used throughout the paper and some
> parts may be confusing and need to be rewritten and clarified. We also agree
> that some related work on paraphrase generation under sampling scenarios is
> missing from the paper and should be added and discussed.

---

### Official Review · Reviewer_dJea · 2025-10-27

**Soundness:** 2
**Presentation:** 2
**Contribution:** 1
**Rating:** 2
**Confidence:** 3

**Summary:**

The paper performs an empirical study of various LLMs on their paraphrase generation capacities. The paper explores multiple hyperparameters - such as decoding style, few shot prompting, vulgar vs standard examples, repetition penalties, and so on.

**Strengths:**

* The studies conduction do provide some insights of effect of some hyperparameters on paraphrase generation.

**Weaknesses:**

1. Overall the study feels incremental - building on an already existing paper [1]. In other words, the paper seems to mainly follow the framework and settings of [1] but explores a couple of hyperparameter variations. The results are also not particularly surprising - such as repetition penalty can increase diversity.

2. Terminology seems inconsistent - especially in evaluation. For example: in the beginning "Sentence production rate" and "Meaning preservation" is mentioned. The later section mentions the use of ParaPLUIE. Then this is not mentioned anymore. Harmonic score mentions "Para" and "Scheme" without clearly linking back to whether it is talking about ParaPLUIE and sentence production rate.

3. Unclear how well the proposed harmonic overall score reflects human judgment.

4. Scheme adherence evaluation seems a bit superfluous. It seems like mainly a matter of adhering to a format like "x) [paraphrase]". This doesn't really seem like a real problem for modern LLMs and tools, as one can do structured generation using outlines/instructor etc. and/or use retries. But in this case, it's not even clear if adherance to structure is all necessary here.

5. I am unclear on the effectiveness or meaningfulness in using an LLM-generated paraphrase corpora in evaluating LLMs for paraphrasing. If we are at the point where LLM-generated corpus can be treated as ground truth - shouldn't this be considered essentially a solved problem? What's a selling point here? We can just use the same strategy as was used for generating the ground truth in practice then.

6. Unclear why no penalty setting uses beam size 1, and others beam size 5. Saying decoding strategy as greedy and beam size as 5 sounds like a contradiction because greedy decoding is beam size with 1 effectively. Moreover, the study seems to miss out the most basic strategy for increasing diversity - i.e using some temperature. Beam decoding is known for lower diversity, and greedy search is effectively temperature 0.

[1] Paraphrase Generation Evaluation Powered by an LLM:
A Semantic Metric, Not a Lexical One

**Questions:**

Why use different beam size for no penalty vs small/high penalty?

---

> ### Author Response · Authors · 2025-11-20
>
> Thanks for your review and your proofreading.
>
> This paper is not a follow-up of [1]. [1] proposes a semantic metric and evaluates it. In this paper we focus on the ability of LLMs to generate paraphrases
> and indeed use the metric proposed in [1].
>
> Regarding the Harmonic score, ”Para” is the rate of generations detected
> as paraphrases with respect to the ParaPLUIE metric. ”Scheme” is the rate
> of generations that have successfully followed the output scheme, which is the
> sentence production rate.
> We used the LLM generated corpus as input sentences. We never considered
> pairs in the corpus as source and ground-truth paraphrase. It was only used as
> an input dataset.
>
> This corpus has been hand-checked in [1], which showed that LMs used
> for generations performed well overall but was not perfect, as it contained 79
> percent of paraphrases. This means that 21 percent of the time,the generation
> is not a paraphrase of the source.
>
> Indeed, modern gargantuan LLMs are able to follow a generation scheme
> however it is difficult for medium-sized LLMs, so in this setup it remain challenging.
> Considering the computational cost of generating with models of the
> size of chatGPT or DeepSeek, this factor should also be taken into consideration.
>
> We agree that the paper lacks clarity regarding the details of beam usage.
> We used 5 groups with each a beam of 1, and this should be explicitly mentioned.
> We choose this setup to take advantage of the DBS algorithm proposed in [2],
> which applies a diversity penalty between beam groups. A diversity penalty
> discourages candidate tokens that have already appeared in other beam groups
> [2]. This becomes relevant in a greedy decoding strategies and allows comparison
> with decoding without diversity penalty. To sum up, with the diversity penalty,
> the decoding compares 5 groups with each a beam of 1, whereas in the no
> penalty setup, there is only one group with a beam of 1.
> The use of beams in a greedy context is also relevant, as it allows comparison
> of sequence perplexity while remaining deterministic. It is important to note
> that the results of a generation in greedy decoding without beams (equivalent
> to beam 1) differ from a generation with a larger beam. In this experiment, all
> beams are of size 1 to ensure consistent comparisons.
>
> We hope that our answers clarify your questions. We agree that this work
> lacks novelty and that there is a need for coherence in the terminology used
> throughout the paper.
>
> [1] Paraphrase Generation Evaluation Powered by an LLM: A Semantic Metric, Not a Lexical One
>
> [2] Diverse Beam Search: Decoding Diverse Solutions From Neural Sequence Models

---

> > ### Comment · Reviewer_dJea · 2025-11-26
> >
> > I appreciate the clarifications. Overall I maintain the current score given the corcerns of novelty and presentational polishing remains.
> >
> > Some additional feedbacks:
> >
> > * Regarding "follow-up on [1]", what I tried to meant was that many of the foundational elements in the current paper seems to be already introduced in [1] - example the datasets chosen for experiments, toxic prompts, the ParaPLUIE metric etc. So it seens like a further hyperparameter exploration on that settings. This in itself is nothing "wrong" per se, it just limits the novelty for ICLR given much of what may seem a bit new in the current paper already exists in [1]. That said it is not that hyperparameter exploration is not desirable in and of itself in a top-tier conference but there should be a stronger selling point discovery or some unique exploratory direction.
> >
> > > This corpus has been hand-checked in [1], which showed that LMs used for generations performed well overall but was not perfect, as it contained 79 percent of paraphrases. This means that 21 percent of the time,the generation is not a paraphrase of the source.
> >
> > This sounds like a very high error rate - and makes me skeptical if we want to use this corpus; although I suppose, some of the established non-LLM paraphrase corpus are also not as perfect either - involving heuristic methods of mining.
> >
> > But now that, with LLMs, we are (I suppose) approaching the limits for these kind of, arguably "simpler" tasks such as generic paraphrasing -- the quality of corpuse and evaluation may be much more critical to care about.
> >
> > > Indeed, modern gargantuan LLMs are able to follow a generation scheme however it is difficult for medium-sized LLMs, so in this setup it remain challenging. Considering the computational cost of generating with models of the size of chatGPT or DeepSeek, this factor should also be taken into consideration.
> >
> > Regarding this what I was wondering was that - do we even need a schema? I suppose you may need one in a raw completion model, but we generally also have instruction-tuned and chat completion models, which are already trained to follow some specific conversational template quite well. For paraphrase generation, paraphrases can be directly responded by the "assistant" role in the conversational API or its equivalent, without any paraphrase-specific schema.

---

### Official Review · Reviewer_1J2K · 2025-10-30

**Soundness:** 2
**Presentation:** 3
**Contribution:** 2
**Rating:** 4
**Confidence:** 5

**Summary:**

**Review Summary of 'Few-Shot Paraphrase Generation with LLMs: An Empirical Study of Models and Hyperparameters'**

This paper offers a detailed empirical evaluation of LLMs on paraphrase generation, with a particular focus on maximizing lexical diversity while maintaining meaning. It compares two types of models: small fine-tuned paraphrasers (e.g., BART, T5) and medium-sized instruction-tuned chat models (3B–8B), tested across four curated datasets. Each model is evaluated with varying prompting schemes (zero-, one-, and four-shot), two prompt styles (standard vs. vulgar), and two output formats (free vs. continuation-style). A total of 10 prompting templates are defined.

The decoding strategy uses beam search only, with three penalty settings (none, small, huge) to promote diversity, notably omitting stochastic sampling. Evaluation spans three axes: format adherence, semantic fidelity (via ParaPLUIE), and lexical diversity (edit and Jaccard distance).

Key findings include: (1) prefix-guided output formatting significantly boosts adherence, (2) diversity penalties increase lexical diversity without heavily sacrificing meaning, and (3) few-shot prompting often reduces diversity—likely due to overfitting to example styles. These trends are robust across model types and datasets.

However, the work's novelty is modest. The use of penalties and prompt variants confirms known behaviors, and the exclusion of larger-scale models (e.g., GPT-3.5/4) limits generality. The absence of sampling-based decoding is a methodological gap, as it’s a common tool for increasing output diversity. Additionally, the inclusion of vulgar prompts is only lightly justified and contributes little to the core goals.

While the methodology is systematic and consistent across models, the evaluation lacks deeper qualitative analysis and human validation. The paper’s structure and writing are adequate but occasionally unclear, especially regarding prompt setup and model formatting. Overall, the study provides a useful, though incremental, benchmark on prompt and decoding effects for LLM-based paraphrasing, with practical insights into how format constraints and decoding knobs affect paraphrase diversity and fidelity.

**Strengths:**

**Strengths**

1. **Thorough Experimental Design**: The study explores a wide range of prompting strategies and decoding configurations across several instruction-tuned and fine-tuned models, enabling consistent comparisons.

2. **Focus on Lexical Diversity**: Emphasizing lexical variation is a valuable angle for tasks like data augmentation, and metrics like edit and Jaccard distance provide useful indicators.

3. **Effective Format Control**: The continuation prompting strategy notably improves adherence to output templates, simplifying evaluation and increasing reliability.

4. **Insightful Prompting Observation**: The paper identifies that few-shot prompting can reduce diversity, a counterintuitive but practically relevant finding.

5. **Use of Automatic Semantic Metric**: Leveraging ParaPLUIE for meaning preservation allows for scalable semantic evaluation beyond overlap-based metrics.

6. **Reproducibility and Scope**: Testing publicly available models across over 200 outputs per input and applying a consistent evaluation protocol adds credibility to the results.

While the contribution is largely empirical and incremental, it offers actionable insights into how prompting and decoding strategies impact paraphrasing performance.

**Weaknesses:**

1. Lack of Novel Methodology**: The paper presents no new model or algorithmic innovation. Its contribution lies primarily in benchmarking existing models with prompt and decoding variants.

2. Omission of Sampling-Based Decoding**: The study excludes common stochastic decoding methods such as top-*k* or nucleus sampling, despite focusing on diversity. This limits the completeness and applicability of the findings.

3. Limited Model Range: Only mid-sized models (up to 8B) are evaluated, with no comparison to larger instruction-tuned models like GPT-3.5 or GPT-4. This restricts generalizability.

4. Superficial Result Analysis: The evaluation relies solely on automated metrics, without qualitative error analysis or human evaluation to validate semantic judgments.

5. Questionable Prompt Variant (Vulgar Examples)**: The inclusion of vulgar prompts is poorly motivated and adds little to the core insights. Its relevance remains unclear.

6. Format Dependency for Evaluation: Strict output formatting is enforced to facilitate automated scoring, but this design choice may penalize otherwise valid paraphrases and obscure natural generation behaviors.

**Questions:**

1. What was the rationale for excluding sampling-based decoding methods, especially given the goal of maximizing paraphrastic diversity?
2. Were the few-shot examples static across all inputs? If so, did you consider using in-context examples closer in domain or structure to the test inputs?
3. How consistent are the findings across datasets? Did any datasets show contradicting trends (e.g., few-shot prompting improving diversity)?
4. What prompted the inclusion of vulgar prompts? Could you clarify what insights, if any, this condition provided relative to the study’s main goals?
5. Did you conduct any human evaluation or spot checks to confirm the accuracy of ParaPLUIE classifications, particularly for borderline or ambiguous paraphrases?
6. How do you anticipate the results might change with larger models or state-of-the-art APIs (e.g., GPT-4)?
7. For practitioners aiming to generate diverse paraphrases, what specific prompting and decoding configurations would you recommend based on your findings?

---

> ### Author Response · Authors · 2025-11-20
>
> Thanks for your review and your proofreading.
>
> 1. We chose to exclude sampling-based decoding strategies as they lack control. Because these strategies rely on randomness and therefore require
> multiple generations in order to create a satisfying candidate. In this work
> we try to focus on a trade-off that creates diversity in a greedy generation
> setting. Thus it could leverage the intense computational cost of LLMs.
>
> 2. The few-shots examples were static across all inputs. We chose to not
> consider in-context examples. The goal here was to observe behaviour of
> models with taboo prompts and the impact of the lexical diversity of the
> shots on the generations. As explained in section 4.3, this seems to have no
> impact on the generation, as expected few shots tends to help the model
> to follow an output scheme but models didn’t catch the lexical diversity
> in the examples provided, which seems an interesting result. We assume
> that using other few-shots examples would lead to the same observation
> considering the wide variety of source sentences used.
>
> 3. We didn’t observe contradictory or different trends across the datasets.
> The graphs were removed at the last minute, mainly because the lack of
> space in the main paper prevented us from discussing them. We can easily
> reintroduce them.
>
> 4. The inclusion of vulgar prompts allows us to confirm that shots have a
> marginal impact on the diversity of the generated outputs. We draw this
> conclusion from the fact that vulgar examples and standard ones didn’t
> share the same lexical diversity (vulgar examples contains mostly different
> words were standard are closer). We would like to observe if the use of
> vulgarity disturbs LLMs and lead to diverse generations. However both of
> these aspects seem to be marginally useful as highlighted by the Appendix 4. figure 5 and Appendix 5. figure 6.
>
> 5. We conducted human evaluation on the sentence pairs identified by our
> post-verification pipeline described in Meaning preservation, Section 3.2. We also identified sentences that were not successfully paraphrased with
> respect to our oracle to gain more insight into language models behavior,
> as detailed in Appendix 2.
>
> 6. It is likely that state of art models would be better in consistency and
> diversity as they contain more knowledge. However, it would be interesting to compare them as well as we observe that very small models such
> as T5-chatGPt-paraphraser could be competitive. This model, trained on
> OpenAI’s gpt-3.5-turbo outputs provides a hint on the quality of chatGPT
> generations as it achieves competitive performance. Therefore, we don’t
> want to include chatGPT in this study due to reproducibility issues. We
> don’t know the architecture behind the API provide by OpenAI and the
> model could change between experiment sessions. Moreover, this experiment already took more than 1500 hours of computation on MI300 device
> provided by Adastra’s calculator.
>
> 7. Based on our findings, we would recommend using a medium-sized model
> in a continuous prompting setting with huge penalties in the decoding
> parameters. We observe that it allows the generation of paraphrases that
> are lexically distant from the original source sentence while maintaining
> adherence to the required scheme, which permits the construction of a
> lightweight automatic paraphrasing pipeline. Moreover, if trained with
> quality data, small models appear to be a good alternative to reduce the
> computational requirements of language models.
>
> We hope that our answers clarify your questions.

---

### Author Response · Authors · 2025-11-20

Thanks for your reviews that will help us enhance the paper.

We agree that the paper is confusing in some parts and need to be clarified.
We also agree on the fact that some references should be added and discussed.
We understand your point of view about the sampling strategy and it should be included in our
experiments for comparison.

---

### Meta-Review · Area_Chair_nKAG · 2026-01-01

**Summary:**

None of the reviewers appears to be advocating for this work.

The authors have acknowledged the need to clarify the confusing aspects of the paper, including references, sampling strategy, and comparative experiments.

**Reviewer Concerns:**

Reviewer 1J2K: (1) limited decoding strategy; (2) key findings are likely due to overfitting to example styles, (3) novelty is modest, and (4) the evaluation lacks deeper qualitative analysis and human validation
*(AC: most concerns are still outstanding)*

---
Reviewer dJea: (1) the study feels incremental; (2) Terminology seems inconsistent; (3) Unclear how well the proposed harmonic overall score reflects human judgment. (4) Scheme adherence evaluation seems a bit superfluous. (5) unclear on the effectiveness or meaningfulness in using an LLM-generated paraphrase corpora in evaluating LLMs for paraphrasing (6) questioning the setting that uses beam size 1, and others beam size 5.
*(AC: most concerns are still outstanding)*

---
Reviewer g2GD: (1) The experimental design for "multiple generation" is logically inconsistent; (2) construction of the overall indicators 𝐻_{𝑚, 𝑝, 𝑑} lacks theoretical support; (3) For the semantic fidelity assessment, the paper relies entirely on the LLM judge, ParaPLUIE, which introduces potential common bias; (4) The conclusion that "few-shot prompts reduce diversity" also needs to be interpreted with caution. (5) The LLM dataset used in the paper is potentially contaminated. (6) It is suggested that there is an inherent coupling problem between the template and the format adherence rate.
*(AC: most concerns are still outstanding)*

---
Reviewer 5cT3: (1) Limited novelty of the paper; (2) Choice of metrics; (3) Missing related work; (4) Readability of the figures/paper as a whole.
*(AC: most concerns are still outstanding)*

**Reviewer Scores:**

Reviewer 1J2K *is likely to keep the score*.
Reviewer dJea *replied and will keep the score*.
Reviewer g2GD *is likely to keep the score*.
Reviewer 5cT3 *is likely to keep the score*.

---

### Decision · Program_Chairs · 2026-01-26

Reject